# Molecular Dynamics Simulations Based on 1-Phenyl-4-Benzoyl-1-Hydro-Triazole ERRα Inverse Agonists

**DOI:** 10.3390/ijms22073724

**Published:** 2021-04-02

**Authors:** Zhipei Gao, Yongli Du, Xiehuang Sheng, Jingkang Shen

**Affiliations:** 1School of Chemistry and Chemical Engineering, Qilu University of Technology (Shandong Academy of Sciences), 3501 Da Xue Road, Jinan 250353, China; 1043117123@stu.qlu.edu.cn; 2College of Chemistry, Chemical Engineering and Materials Science, Shandong Normal University, 88 Wen Hua Dong Road, Jinan 250014, China; 3State Key Laboratory of Drug Research, Shanghai Institute of Materia Medica, Chinese Academy of Sciences, 555 Zu Chong Zhi Road, Shanghai 201203, China; jkshen@mail.shcnc.ac.cn

**Keywords:** ERRα, inverse agonists, 1-phenyl-4-benzoyl-1-hydro-triazole, MD simulations

## Abstract

Estrogen-related receptor α (ERRα), which is overexpressed in a variety of cancers has been considered as an effective target for anticancer therapy. ERRα inverse agonists have been proven to effectively inhibit the migration and invasion of cancer cells. As few crystalline complexes have been reported, molecular dynamics (MD) simulations were carried out in this study to deepen the understanding of the interaction mechanism between inverse agonists and ERRα. The binding free energy was analyzed by the MM-GBSA method. The results show that the total binding free energy was positively correlated with the biological activity of an inverse agonist. The interaction of the inverse agonist with the hydrophobic interlayer composed of Phe328 and Phe495 had an important impact on the biological activity of inverse agonists, which was confirmed by the decomposition of energy on residues. As Glu331 flipped and formed a hydrogen bond with Arg372 in the MD simulation process, the formation of hydrogen bond interaction with Glu331 was not a necessary condition for the compound to act as an inverse agonist. These rules provide guidance for the design of new inverse agonists.

## 1. Introduction

Estrogen-related receptor α (ERRα) was the first subtype identified in the family of estrogen-related receptors [1], which play an important role in the expression of oxidative metabolism genes and mitochondrial biosynthesis by interacting with oxidative metabolism transcription genes PGC-1α and RIP140 [2,3]. PGC-1α can induce the expression of ERRα and activate the transcription of ERRα by physically binding with ERRα [4]. Experiments have shown that ERRα is overexpressed in the incidence, progression, and poor prognosis of a variety of cancers, such as breast [5], endometrial [6], prostate [7], ovarian [8], and colon cancers [9]. In breast cancer cells, ERRα could regulate the level of estrogen receptor α (ERα) gene transcription by competing to bind estrogen response elements with ERα [10]. A study of breast cancer cells in xenograft models indicated that knocking out the ERRα gene could delay the growth of cancer cells [11]. In vivo studies have shown that inverse agonists of ERRα have a general inhibitory activity on the proliferation of breast cancer cells, including hormone-dependent and non-hormone-dependent breast cancer cells [12]. In addition, turning down the expression of ERRα in the MDA-MB-231 cell line could inhibit the migration activity of the MDA-MB-231 cell line [13], indicating that converting the apoERRα to an inactive state may help prevent the metastasis of tumor cells. ERRα plays a regulatory role in the growth and invasion of hormone-dependent and non-hormone-dependent breast cancer cells. ERRα could regulate the proliferation of vascular smooth muscle cells through the protein kinase ERK signaling pathway [12] and regulate the formation of endothelial microtubules and tumor angiogenesis by adjusting the growth of VEGF expression and PI3K/Akt/STAT3 signaling pathway [14]. Inverse agonists could inhibit the proliferation and metastasis of tumor cells by turning down the expression of ERRα. Thus, the development of effective ERRα selective inverse agonists may be effective treatment for breast cancer.

The crystal structure shows that ERRα is a sandwich structure composed of 12 α-helixes and 2 β-sheets [15]. The comparison between the crystal structure of apoERRα (PDB code: 1XB7) [15] and inverse active state (PDB code: 2PJL) [16] shows that after the inverse agonist binds to ERRα, the ligand binding domain undergoes major structural changes to accommodate the inverse agonist. The indole part of the inverse agonist compound **1** (ligand of crystal structure 2PJL) occupies the position of residue Leu324 on Helix3, while the indole amino group of compound **1** occupies the position of residue Phe328 on Helix3, causing Helix3 to stretch out. Residue Phe328 pushes residue Phe510 on Helix12, which in turn pushes Helix12 to flip, occupying the binding site of PGC-1α in apoERRα, so that PGC-1α cannot continue to activate the transcription of ERRα, and ERRα assumes an inactive conformation (Figure 1). The nitrogen atom on the amino group of compound **1** forms a hydrogen bond interaction with the Glu331 on Helix3, and the other hydrophobic parts on compound **1** form a hydrophobic interaction with the hydrophobic groups (Phe328, Phe382, Leu398, Phe495, and Leu500) that constitute the ligand binding domain. The study found that the changes that had important contributions to improving the inverse agonist activity have significance in guiding the design of new inverse agonists.

In recent years, several series of ERRα small molecule inverse agonists have been reported (shown in Figure 2). Thiadiazole acrylamide (XCT-790) was the first selective and effective ERRα inverse agonist reported in the literature, and its IC_50_ value was 0.37 μM. However, the crystal complex has not been reported [17]. ERRα inverse agonists based on N-arylindole (compound **1**) were the first reported crystalline complex of inverse agonist with ERRα [16]. Compound **1** had an IC_50_ value of 0.19 μM in cells and was slightly more potent than XCT-790. A series of ERRα inverse agonists based on di-aryl ether thiazolidinedione (compound **2**) were reported in 2011, with the best biological activity at the molecular level (IC_50_ = 0.04 μM). This was the first reported crystalline complex with covalent bonding. In 2017, Patch optimized its di-aryl ether part and obtained compound **3** with an in vitro IC_50_ of 0.008 μM [18,19]. In 2013, a series of ERRα inverse agonists based on 1-phenyl-4-benzoyl-1-hydro-triazole, with the best biological activity in vitro of 0.021 ± 0.009 μM, were reported (compounds **4**, **5**, and **6**) [20]; they had the characteristics of simple structure and high activity, and pharmacokinetic studies showed that they had good oral bioavailability. In 2016–2017, two series of compounds **8**, **9**, and **10** with micromolar biological activity at cell and animal levels were reported, and compound **10** had the best biological activity among them with an IC_50_ of 0.64 ± 0.12 μM in MDA-MB-231 cells [21,22].

As there have been few reports on crystalline complexes, there is still a need for in-depth study of the binding mode of ERRα with inverse agonists, the amino acid residues that play a key role, and the relationship between the structure of inverse agonists and their activities. The structure–activity relationship analysis of compounds **4**, **5**, and **6** and the same series of compounds found that simple modifications in the structure could cause major changes in biological activity. Among them, compounds **4**, **5**, and **6** had the characteristics of simple structure and strong biological activity. In order to study the mode and law of the structural relationship of inverse agonists with ERRα in depth, a molecular dynamics simulation study of compounds **4**, **5**, and **6** was carried out. The results presented in this article may provide guidance for the design of new ERRα inverse agonists.

## 2. Materials and Methods

### 2.1. Protein Preparation and Molecular Docking

The dynamic structure of ERRα reverse excitation was obtained from the Protein Data Bank (PDB code: 2PJL) [16]. The missing residues (309–317 and 462–470) were completed by the SWISS-MODEL server [23], and the ERRα protein was protonated by PDB2PQR [24]. ERRα small molecule inverse agonist compounds **4**, **5**, and **6** were selected as the research objects, and the ligand binding domain (LBD) of ERRα was characterized by the fragment-centric topographic mapping tool AlphaSpace. In order to obtain the conformational state of the inverse agonist compounds **4**, **5**, and **6** and ERRα with the lowest binding energy as the initial state of the molecular dynamics simulation, Sybyl-x2.0 was used to dock the compounds **4**, **5**, and **6** to the ERRα LBD, and a binding model of inverse agonists and ERRα was selected according to the spatial extent of the active pocket. 

### 2.2. Molecular Dynamics Simulation

AMBER14 software package was used for molecular dynamics simulation [25]; the target ERRα was processed by ff14SB force field [26]; and for inverse agonist compounds **4**, **5**, and **6,** GAFF force field was used to parameterize bonding and van der Waals parameters. For inverse agonist compounds **4**, **5**, and **6** at the level of B3LYP/6-31+G(d,p), GAUSSIAN 09 was used for structural optimization [27], and the RESP method was used to fit the charge to each atomic center [28]. Through the tleap module in AMBERtools 17, an octahedral water box with a margin of 12 Å was used, and a Cl^–^ neutralization system was added. In the simulation process, after the energy of all systems was minimized, the system was heated from 0 to 300 K under NPT ensemble conditions. After the 300 K system reached equilibrium for 2 ns, molecular dynamics simulation was performed under NPT ensemble conditions for 1.0 μs, the track coordinates were saved every 2 ps, and the rest were calculated according to the default parameter settings of the software itself. The cpptraj module [29] in AMBERtools 17 was used to analyze the simulated conventional parameters RMSD, distance, cluster analysis, etc. Hbond command in AMBERtools 17 was used to identify the hydrogen bond interaction between compound and protein, whose geometric standard was distance within 3.0 Å and an angle cutoff of 135°. Graphical analysis was performed with Chimera software.

### 2.3. Binding Free Energy Calculation

The MM/GBSA method was used to calculate the binding free energy of inverse agonist compounds **4**, **5**, and **6** with ERRα [30]. For each complex, a total of 50 snapshots were extracted along the molecular dynamics (MD) trajectory from the last 100 ns MD simulations with an interval of 200 ps. The binding energy (Δ*G*) in condensed phase can be simply defined by the following equation:$$\Delta G=\;\Delta E_{ele}+\;\Delta E_{vdw}+\Delta G_{pol}+\;\Delta G_{nonpol}\;-T\Delta S$$
in which the first two terms (Δ*E_ele_* and Δ*E_vdw_*) are electrostatic and van der Waals interactions in the gas phase, respectively, and are usually computed using molecular mechanics. The term Δ*G_pol_* represents the polar solvation free energy that can be obtained by solving the Poisson–Boltzmann equation. The term Δ*G_nonpol_* is the nonpolar solvation free energy and is computed by the following empirical equation: $$\Delta G_{nonpol}=\;\gamma \times SASA+\;\beta$$
where the values of *γ* and *β* are set as 0.00542 kcal/mol A^−2^ and 0.92 kcal/mol [31], respectively. The atom radii in the prmtop files were set as the default values. The dielectric constants for the solute and surrounding solvent are 1.0 and 80.0, respectively. For the last term, the entropy (−*T*Δ*S*) was calculated using classical statistical thermodynamics and normal mode analysis [32].

### 2.4. Clustering Analysis

The clustering of trajectories was performed using cpptraj module in AMBER17. The clustering was done with a minimum number of points and an epsilon value of 4.0, taking the first frame as a reference. The cluster-to-cluster distance was defined as the average of all distances between individual points of two clusters according to the so-called average-linkage algorithm, which is one of the best clustering methods [33]. In total, we generated 10 clusters and considered the closest cluster conformation with reference structure.

## 3. Results and Discussion

In order to deeply explore the combination of ERRα with inverse agonists and guide the design of new inverse agonists, the 1.0 μs molecular dynamics simulations of ERRα with its inverse agonists (compounds **4**, **5**, and **6**) were performed. The structural stability of the complexes was evaluated by observing the RMSD of the main chain atoms. As shown in Figure 3A, the root-mean-square deviation (RMSD) of the combined simulation system of compounds **4** and **5** with ERRα tended to be stable after 200 ns. The 1.0 μs molecular dynamics simulation keeps the system in a relatively stable state. Compound **6** exhibited certain fluctuation characteristics in stable RMSD, probably due to its unique binding mode. The RMSD of the ligand binding domain (LBD) of compounds **4**–**6** with ERRα was calculated to evaluate the stability of the LBD. As shown in Figure 3B, the RMSD of compounds **4** and **5** tended to be stable after 200 ns. Similar to the RMSD of main chain atoms, the unique binding mode of compound **6** may cause its RMSD of the LBD to tend to be stable after 700 ns. 

The analysis of the crystalline complex 2PJL shows that the binding pocket of ERRα inverse agonist is mainly composed of 19 amino acid residues within 4.2 Å from the inverse agonist (Figure 4), including H3 (Val321, Leu324, Leu327, Phe328, Glu331), H5 (Leu365, Val366, Val369), *β*-sheet (Phe382, Leu386, Leu388), H6 (Ala396), the H6/H7 loop area (Gly397, Leu398), H7 (Leu405), H11 (Val491, Phe495, Val498), and H11/H12 loop regions (Leu500). As the ligand binding domains were mainly composed of the hydrophobic amino acids, the major interactions between inverse agonists and ERRα were hydrophobic interactions.

In the MD simulation results of compound **4** with ERRα (Figure 5A), residue Glu331 rotated and formed a stable hydrogen bond with the Arg372. Compared with the crystalline complex 2PJL, the complex of compound **4** with ERRα shows that the front part of Helix3 tightened toward the LBD, which caused the rotated residue Glu331 located on Helix3 to form a strong hydrogen bond (distance 1.871 Å) with the amino group on compound **4**. Moreover, the methyl group on compound **4** collided with Phe495 on Helix11, which caused the terminal part of Helix11 to unwind and move out of the ligand binding domain. Phe328, Glu331, and Phe495 extended outward to accommodate the binding of compound **4**. 

Similar to compound **4**, the Glu331 in the MD simulation result of compound **5** (Figure 6A) also rotated and formed a stable hydrogen bond with Arg372. Compared with the crystalline complex 2PJL, the MD simulation results of compound **5** show that the anterior segment of Helix3 was more inclined to the LBD, which caused the Ser325 located on Helix3 to form a strong hydrogen bond (distance 1.861 Å) with the amino group on compound **5**. Moreover, the methyl group on compound **5** collided with Phe495 on Helix11, which caused the posterior segment of Helix11 to uncoil further and become more inclined to the LBD. In addition, the extension of the aniline group in the ligand binding pocket on compound **5** caused the front segment of Helix12 to stretch out of the ligand binding domain. The 1H-1,2,3-triazole moieties on compound **5** were sandwiched in a hydrophobic interlayer composed of Phe328 and Phe495, while compound **4** was without this interlayer. 

In the molecular dynamics (MD) simulation results of compound **6** with ERRα (Figure 7A), Glu331 formed a stable hydrogen bond with Arg372 after rotation. Similar to compound **5**, 1H-1,2,3-triazole on compound **6** formed a hydrogen bond with Ser325 (distance 2.286 Å). Compared with the crystalline complex 2PJL, in the complex of compound **6** with ERRα, the anterior segment of Helix3 still shrank in the direction of LBD, which caused Ser325 on Helix3 to form a hydrogen bond with 1H-1,2,3-triazole on compound **6**. Moreover, the isopropyl phenyl group on compound **6** collided with Phe495, causing the posterior segment of Helix11 to partially unscrew and stretch out of the ligand binding domain. Besides, the anterior segment of Helix12 extended outward, and the second half of Helix12 tightened towards the LBD. In addition to being sandwiched in the hydrophobic interlayer composed of Phe328 and Phe495 like compound **5**, the aniline group on compound **6** was also sandwiched in the hydrophobic interlayer composed of Phe382 and Phe495.

The MD simulation results of compound **4** with ERRα (Table 1) show that there was a strong interaction between the residues Phe328 and Leu365 and compound **4**, whose binding free energy was less than −3, which made a greater contribution to the binding activity. The binding free energy between Glu331 and compound **4** was less than −2, which also contributed to biological activity. Val491, Leu401, Phe382, Val369, and Met362 with compound 4 also have binding free energy of less than −1. The MD simulation results of compound **5** with ERRα (Table 1) show that the interaction between Phe328 and compound **5** was the strongest (Δ*G_total_* < −3), which contributed greatly to the binding activity. At the same time, Ser325, Phe495, Lys508, Leu324, and Asp329 with compound 5 also had binding free energy of less than −2, which also contributed to the potent binding activity. The interaction between compound **6** and residue Phe328 was particularly strong (Table 1), and the binding free energy was less than −4.7, which contributed the most to the binding activity, suggesting that Phe328 residue had a decisive influence on the activity of inverse agonists. The interaction between residue Phe495 and compound **6** was also very strong, and the contribution of binding free energy was less than −3.2, suggesting that Phe495 has an important influence on the strong activity of inverse agonists. The contribution of Leu324 to the binding free energy was less than −2.3, and the contributions of Leu398, Phe382, Ser325, Pro505, and Lys503 to the free energy of binding were all less than −1, which made a greater contribution to the activity of the inverse agonist compound **6**. Surprisingly, the total binding energies of compounds **4**, **5**, and **6** from the residues Phe328 and Phe495 that constituted the hydrophobic interlayer were positively correlated with their biological activities (−2.84151 kcal/mol, −6.00085 kcal/mol, and −7.96174 kcal/mol separately, shown in Table 1). It is suggested that the two residues of Phe328 and Phe495 had a very important influence on the activity of inverse agonists of ERRα and had important guiding significance for the design of novel inverse agonists. The amino group on the weakest compound **4** only formed a hydrogen bond interaction with Glu331, which may be the reason for its weaker activity. Moreover, during the dynamic simulation process, Glu331 flipped and formed a strong hydrogen bond with Arg372, which suggests that Glu331 was relatively unstable and the formation of hydrogen bonds between inverse agonists and Glu331 was not necessary for high biological activity.

The MD simulation results of compounds **4**, **5**, and **6** with ERRα show that Leu324, Phe328, Met362, Leu365, Val366, Phe382, Leu398, Leu401, Val491, and Phe495 were involved in the formation of the ligand binding domain, which indicates that these residues may be the key residues that contribute greatly to the interaction between ERRα and inverse agonists. As most of these residues are hydrophobic residues, the main interactions should be hydrophobic. The total binding energies of compounds **4**, **5**, and **6** with ERRα were calculated, and the results showed that the van der Waals force between the ligands and the protein contributed more to the total binding energy than the electrostatic interactions, which demonstrates that the ligand binding domain of ERRα is a hydrophobic pocket (Table 2).

## 4. Conclusions

To determine the binding mode, key amino acid residues, and total binding free energy of inverse agonists compounds **4**, **5**, and **6** with ERRα, molecular docking and molecular dynamics simulations were carried out. The results of molecular dynamics simulations show that the different binding modes of small molecules in the LBD lead to changes in helix structures, which affects the biological activity of the inverse agonist. The binding mode of small molecules in the LBD caused changes in Helix3 and Helix11. The flexibility of Helix3 and Helix11 is positively associated with the biological activity of the inverse agonist. The interaction between small molecules and ERRα is mainly hydrophobic. Whether a hydrophobic interlayer was formed with Phe328 and Phe495 and whether there was a strong interaction with the residues that constitute the interlayer had important effects on the biological activity of inverse agonists. As Glu331 flipped during the MD simulation process and formed a hydrogen bond with Arg372, the formation of hydrogen bond interaction with Glu331 was not a necessary condition for the compounds to act as inverse agonists. These results can provide guidance for the design of novel potent effective ERRα inverse agonists.

## Figures and Tables

**Figure 1 ijms-22-03724-f001:**
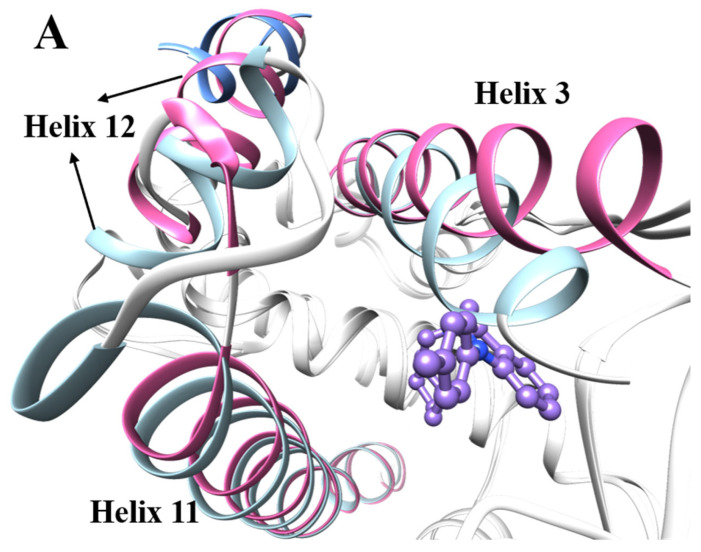
(**A**) Sky blue: helix in crystal structure of apoERRα (PDB code: 1XB7); orchid: helix in the complex of estrogen-related receptor α (ERRα) with inverse agonist (PDB code: 2PJL); medium purple: inverse agonist of crystal structure 2PJL (compound **1**); cornflower blue: PGC-1α in crystal structure of apoERRα. (**B**) Changes of major amino acid residues between apoERRα and inverse state. Sky blue: residues in crystal structure apoERRα (PDB code: 1XB7); orchid: residues in crystal structure of ERRα with inverse agonist (PDB code: 2PJL).

**Figure 2 ijms-22-03724-f002:**
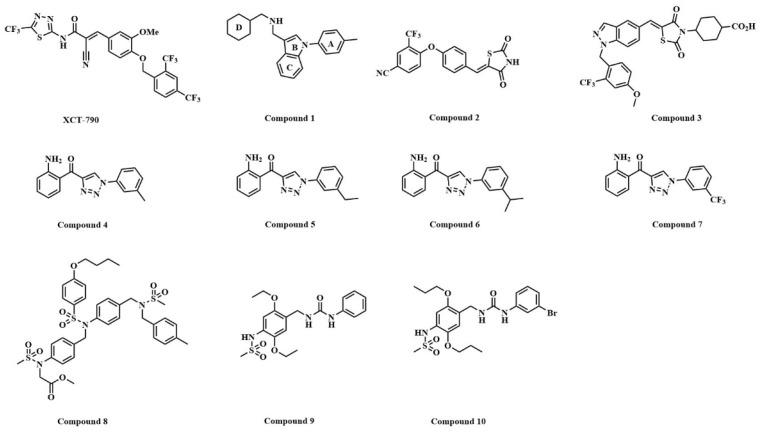
The structure of representative ERRα inverse agonists.

**Figure 3 ijms-22-03724-f003:**
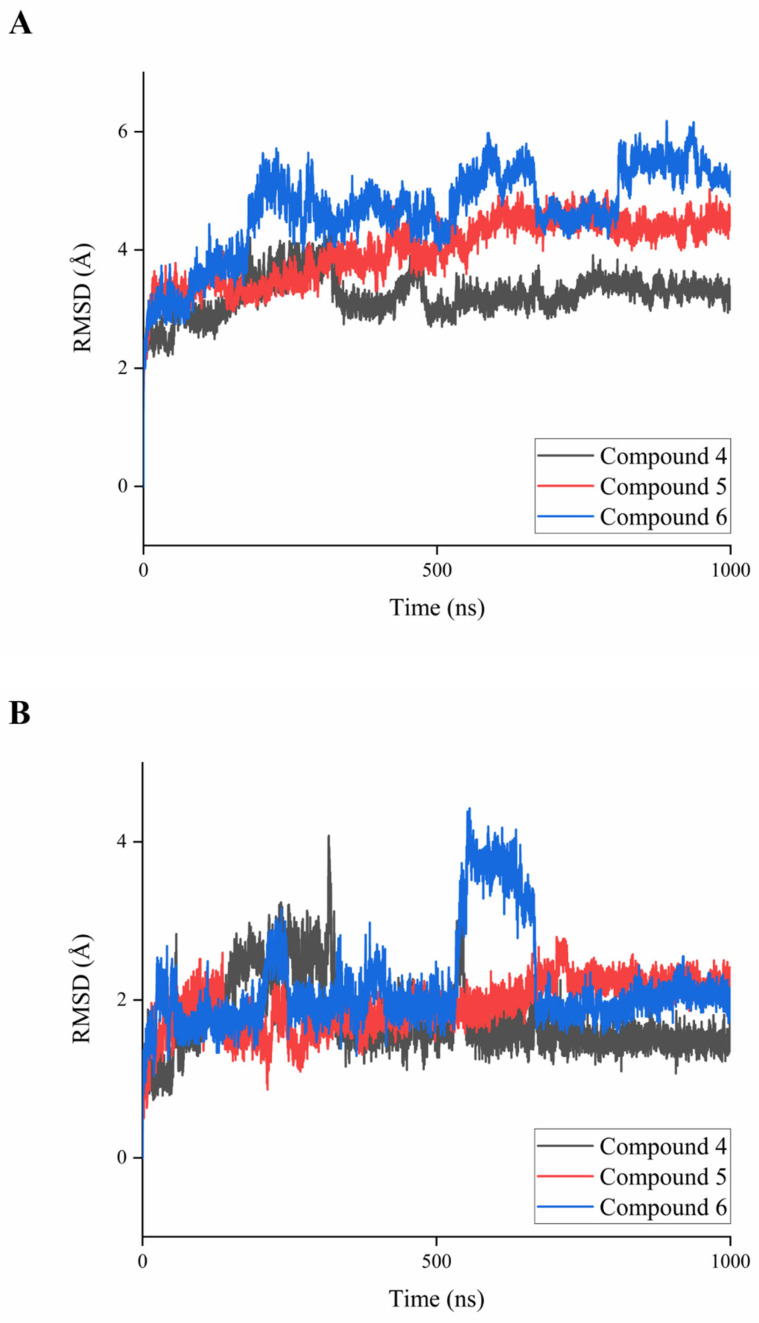
(**A**) RMSD of the 1 μs dynamics simulation for complexes of compounds **4**–**6** with ERRα. (**B**) RMSD of the 1 μs dynamics simulation for ligand binding domain (LBD) of compounds **4**–**6** with ERRα.

**Figure 4 ijms-22-03724-f004:**
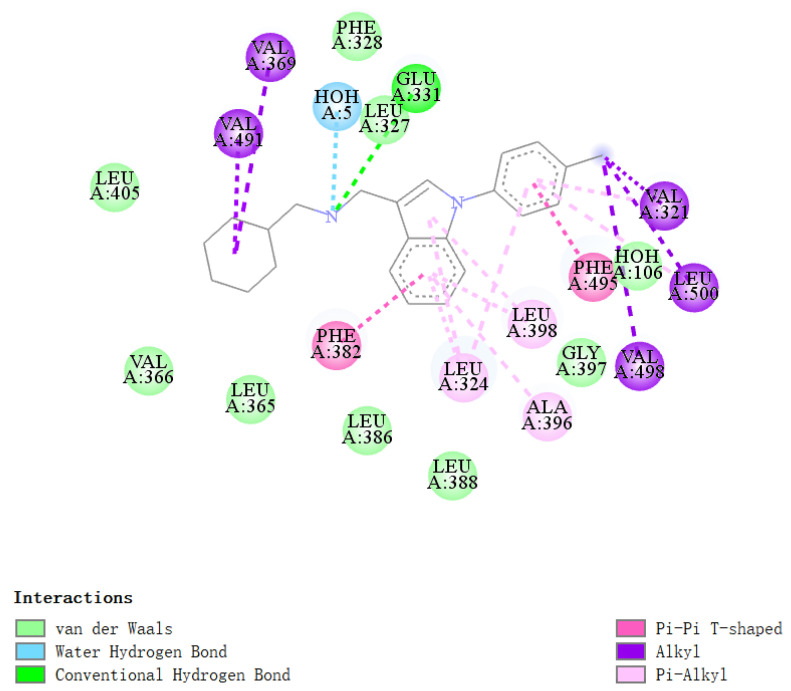
Amino acid residues constituting the LBD of compound **1** with ERRα and interactions between compound **1** and residues of ERRα. This figure was generated with Discovery Studio 2016.

**Figure 5 ijms-22-03724-f005:**
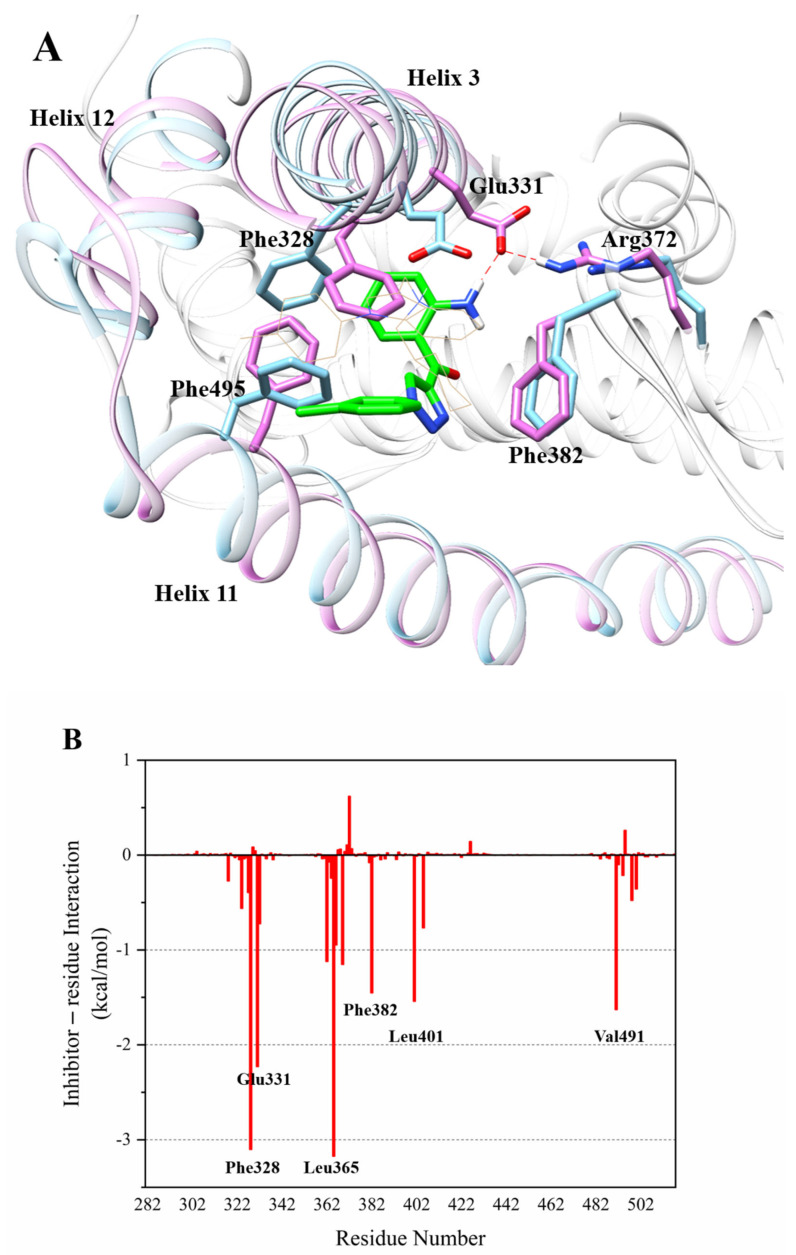
(**A**) Orchid: helixes and residues in complex of compound **4** with ERRα; sky blue: helixes and residues in complex of compound **1** with ERRα (PDB code: 2PJL); green: compound **4**. (**B**) Energy contribution of each amino acid residue in complex of compound **4** with ERRα.

**Figure 6 ijms-22-03724-f006:**
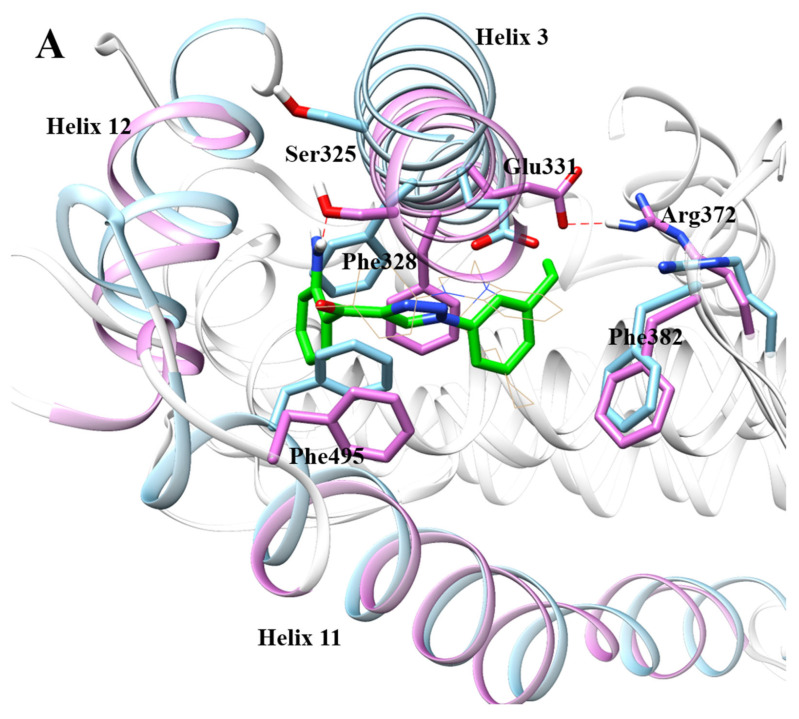
(**A**) Orchid: Helixes and residues in complex of compound **5** with ERRα; sky blue: helixes and residues in complex of compound **1** with ERRα (PDB code: 2PJL); green: compound **5**. (**B**) Energy contribution of each amino acid residue in complex of compound **5** with ERRα.

**Figure 7 ijms-22-03724-f007:**
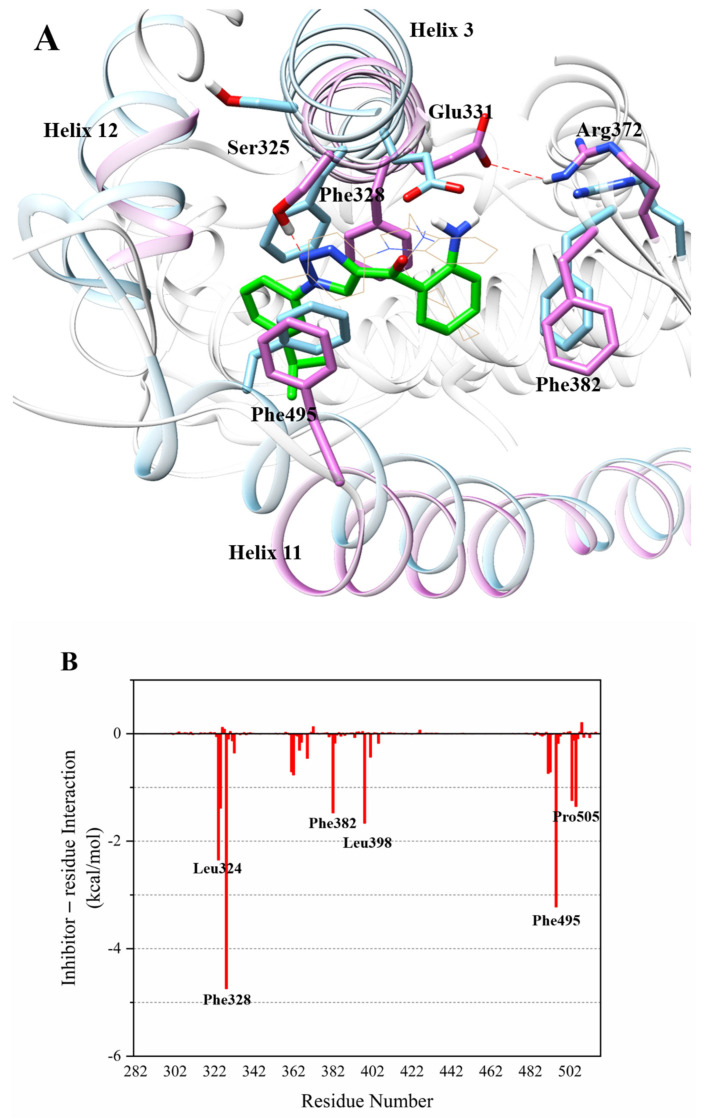
(**A**) Orchid: Helixes and residues in complex of compound **6** with ERRα; sky blue: helixes and residues in complex of compound **1** with ERRα (PDB code: 2PJL); green: compound **6**. (**B**) Energy contribution of each amino acid residue in complex of compound **6** with ERRα.The analysis of the binding mode of compounds **4**, **5**, and **6** with ERRα shows that the main changes in the process of compounds **4**, **5,** and **6** with ERRα were on Helix3 and Helix11, which was also confirmed by the root mean square fluctuation (RMSF) diagram. In the RMSF (Figure 8), as the flexibility of the ERRα protein at Helix3 and Helix11 increases, the biological activities of its corresponding compounds **4**, **5**, and **6** gradually increase, which indicates that the changes of Helix3 and Helix11 have a decisive influence on the activity of inverse agonists. These results also suggest that the inverse agonists form effective hydrogen bonding with Ser325 and form a sandwich with Phe328 and Phe495 through hydrophobic interaction, which would play an important role in enhancing the inverse agonistic activity against ERRα.

**Figure 8 ijms-22-03724-f008:**
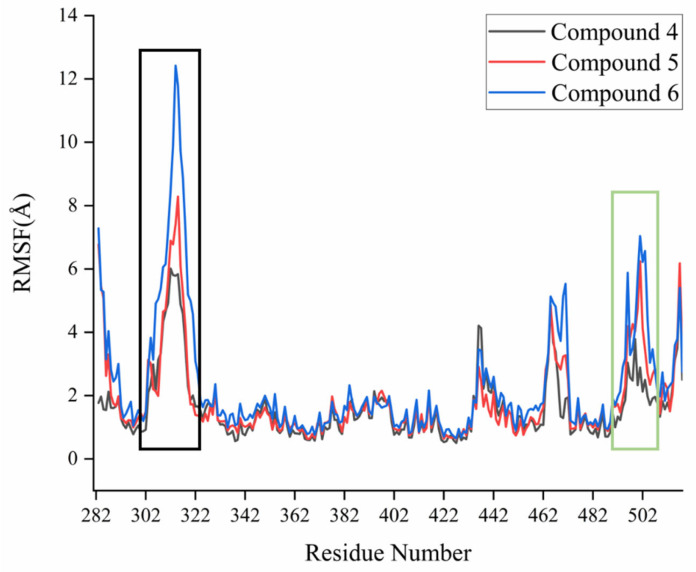
RMSF of ERRα with compounds **4, 5,** and **6**; black frame: residues on Helix 3; green frame: residues on Helix 11.

**Table 1 ijms-22-03724-t001:** Binding free energy decomposition results of important amino acid residues that interact with inverse agonists in ERRα (kcal/mol).

Helix	Residue Number	Compound **4**	Compound **5**	Compound **6**
Helix3	Leu324	−0.56059	−2.5887	−2.34783
Helix3	Ser325	−0.039	−2.95568	−1.38119
Helix3	Leu327	−0.39394	−0.99175	0.079
Helix3	Phe328	−3.10079	−3.13298	−4.74127
Helix3	Asp329	0.082	−2.23817	−0.09419
Helix3	Glu331	−2.22772	−0.40778	−0.126
Helix5	Met362	−1.12072	−0.47778	−0.76175
Helix5	Leu365	−3.17092	−0.55091	−0.30182
Helix5	Val366	−0.94607	−0.21529	−0.15418
Helix5	Val369	−1.15165	−0.34102	−0.45299
β-Sheet	Phe382	−1.45135	−0.93137	−1.46803
H6/H7loop	Leu398	0.001042	−0.70014	−1.66316
Helix7	Leu401	−1.54059	−0.47798	−0.43139
Helix11	Val491	−1.62795	−1.10149	−0.73375
Helix11	Leu492	−0.099	−0.196	−0.7076
Helix11	Phe495	0.259283	−2.86787	−3.22047
Helix11	Val498	−0.47564	−1.1305	−0.004
Helix12	Lys503	0.015	0.05505	−1.23881
Helix12	Pro505	−0.016	−0.085	−1.34973
Helix12	Lys508	0.003	2.72599	0.203202
	Phe328 + Phe495	−2.84151	−6.00085	−7.96174

**Table 2 ijms-22-03724-t002:** Binding thermodynamic data of ERRα with inverse agonists **4**–**6** (kcal/mol).

	Compound **4**	Compound **5**	Compound **6**
Δ*E_vdw_*	−44.0033	−46.8355	−47.989
Δ*E_ele_*	−18.7634	−25.2341	−18.8281
Δ*G_total_*	−39.4772	−42.2583	−44.5491
IC_50_	2.17 ± 0.03 µM	0.20 ± 0.07 µM	0.021 ± 0.009 µM

## Data Availability

Not applicable

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
