# Peer review of "Molecular Dynamics Simulations Based on 1-Phenyl-4-Benzoyl-1-Hydro-Triazole ERRα Inverse Agonists"

_ijms, 2021, doi:10.3390/ijms22073724_

Round 1
Reviewer 1 Report
In this manuscript, Gao and co-workers describe the MD simulations to study the effect of inverse agonists on the structural arrangement and inactivation of ERRα protein. I consider the paper relevant in the field, due to the overexpression of this protein in different cancers.
Some improvements, especially on Figures and captions should be made to improve the quality of the paper and to facilitate the reading and overall comprehension. Below are my detailed comments:
- Figure captions are poor, too simple and lacks important information.
- In Figure 1, the color scheme could be different, cornflower blue > light blue; medium purple > light pink and purple > magenta (pink).
- Quality/resolution of Figures 2, 3, 5, 6, 7 and 8 must be improved.
- The graphics (B) on Figures 5, 6 and 7 are difficult to read.
- How Figure 4 was generated, with the assist of which program/tool?
- Figure 3 caption does not indicate that it is the RMSD of the complex protein::compound. Also, perhaps would be more informative to look at the RMSD of the active site/binding pocket residues, to disclose the effect of the molecules at this site. Consider adding a graphic B with this zoom in.
- In Methods section the analysis must be described in detail. The cluster analysis was used to do what? The distance analysis was used to measure the hydrogen bonds or other distance? What the criteria to define the hydrogen bond between the compound and an amino acid? Only the distance, or also the angle (direction)?
- The MM/GBSA method must be further described in order to allow reproducibility. Which steps are involved? Equations?
Author Response
Dear Reviewer,
We appreciate your comments, and those comments are all valuable and very helpful for revising and improving our paper, as well as the important guiding significance to our researches. We have studied comments carefully and have made correction which we hope meet with approval.
Please see the attachment

Reviewer 2 Report
Authors have carried out a MD study of the interaction mechanism between inverse agonist with ERRa. The binding energy was analysed by using the MM-GBSA method. Authors have carried out well their methodology, there is always a doubt whether the force-field is good enough though. However, author's choice seem rational and there is thus some confidence on their results. The manuscript is well-written and the simulations and analysis have been carefully performed.
All in all, I would like to recommend publication of this manuscript. Of course, the quality of the figures require improvements.
Author Response

(The authors gave the same response as above.)

Round 2
Reviewer 1 Report
The authors satisfactorily responded to many of my concerns, and the manuscript is much more improved.